# Mentalization in Asylum Seekers with Post-Traumatic Stress Disorder

**DOI:** 10.3390/ijerph22091405

**Published:** 2025-09-09

**Authors:** Massimiliano Aragona, Marcella Cavallo, Federica Ferrari, Giovanna Laurendi, Gianluca Nicolella

**Affiliations:** National Institute for Health, Migration and Poverty (INMP), 00153 Rome, Italy; marcella.cavallo@inmp.it (M.C.); federica.ferrari@inmp.it (F.F.); g.laurendi@sanita.it (G.L.); gianluca.nicolella@uniroma1.it (G.N.)

**Keywords:** mentalization, Post-Traumatic Stress Disorder (PTSD), refugees, somatization, alexithymia, empathy

## Abstract

This study explores the relationship between post-traumatic stress disorder (PTSD) and the two major dimensions of mentalization (self and other-oriented) in asylum seekers resettled in Italy. It is important because it is the first study addressing the role of mentalization in asylum seekers with PTSD. Twenty asylum seekers scoring above the cut-off for the PTSD Checklist for DSM-5 (PCL-5) were contrasted to twenty asylum seekers without PTSD on mentalization and somatization. The Certainty About Mental States Questionnaire (CAMSQ) and the Bradford Somatic Inventory (BSI-21) were used to assess mentalization and somatization, respectively. Pearson correlations and multiple regression analyses were conducted to examine the overall association between variables. Asylum seekers with PTSD had significantly higher scores than those without PTSD on somatization (*p* = 0.03), and significantly lower scores on self-oriented mentalization (*p* < 0.05) than those without PTSD. These results show that asylum seekers with PTSD have reduced self-oriented mentalizing abilities, while the other-oriented component of mentalization appears less involved. This study shows that mentalization deserves more research for the possibly crucial role of reduced self-oriented mentalization in asylum seekers’ suffering. If these findings are confirmed by future studies, they could be used to tailor interventions for asylum seekers and refugees with PTSD.

## 1. Introduction

The prevalence of mental distress in asylum seekers and refugees is high, with findings converging on relevant rates of depression, anxiety, somatization, and post-traumatic stress reactions [1,2]. Post-traumatic stress disorder (PTSD) is a severe mental health condition that can develop following exposure to traumatic events. For persons seeking international protection, the risk of developing PTSD is higher due to the deep and often violent experiences they endure, including war, persecution, displacement, and human rights violations [1]. The impact of PTSD on the mental and emotional well-being of international protection applicants is not only a significant public health concern but also poses challenges for their integration and adjustment into host countries. However, PTSD is just one possible reaction to traumas, which may also lead to personality changes, including impaired mentalization [3].

Mentalization is a psychological construct referring to the capability by which human beings, implicitly and/or explicitly, make sense of others and themselves in terms of subjective states and mental processes [4]. Over the past few decades, mentalization has emerged as a central concept in clinical and developmental psychology, and it has increasingly been used as a tool for assessing psychological well-being. Numerous studies [5,6] have shown that good mentalization skills are associated with better emotional, social, and relational functioning. People who can effectively mentalize regulate emotions better; have more satisfying interpersonal relationships; and are less likely to experience disorders such as depression, anxiety, or borderline or narcissistic personality disorders. Therefore, assessing the ability to mentalize can be seen as an indicator of psychological well-being.

Studies show reduced mentalization after adverse childhood experiences [7] and a possible role of mentalization in family dysfunction in veterans [8]. Moreover, impaired mentalization has been suggested in anxiety and related disorders, with specific deficiencies in post-traumatic stress disorder [3], and a meta-analysis found consistent evidence that patients with PTSD score lower than controls on mentalization [9]. However, a study of the same group was unable to demonstrate a mediation role of mentalization for PTSD in veterans [8], suggesting that further evidence is needed to better assess the role of mentalization in the development of PTSD in traumatized individuals.

Among the severely traumatized individuals who often suffer from PTSD and related post-traumatic reactions (e.g., complex PTSD) are asylum seekers, who are exposed to traumatic experiences in their countries and/or during the migration journey. However, to our knowledge, research on the relationship between PTSD and mentalization in asylum seekers is lacking. Consequently, this study aims to fill this gap by exploring the relationship between PTSD and the two major dimensions of mentalization (self and other-oriented).

In our study we will also assess the role of somatization, which refers to the process by which psychological distress is expressed as physical symptoms. We consider somatization because it is frequent among migrants [10], some forms of somatization have a post-traumatic nature [11], and somatization often interacts with PTSD in migrants with traumatic backgrounds [12]. Moreover, mentalization deficits are often associated with somatization [13]. For these reasons, we included somatization in the analysis as a possible relevant confounding covariate to control for.

## 2. Materials and Methods

### 2.1. Participants

In this cross-sectional exploratory study, 40 asylum seekers from the National Institute for Health, Migration and Poverty (INMP) outpatient clinic were enrolled. They were randomly selected among those who had access to our “International Protection Service” in 2024. To enter this study, all subjects signed an individual consent to participate, available in Italian, English, and French. In case of minor linguistic difficulties, a cultural mediator was available to translate the consent and to participate in the person’s native language. Asylum seekers who were unable to read any of these languages were excluded from the study (provided that they could not read the questionnaires (see below)).

Participants were divided into two groups considering whether or not they were affected by PTSD. Sociodemographic variables and clinical information were collected during a routine visit. Inclusion criteria were as follows: being an asylum seeker or refugee; age between 18 and 80 years old; having no impairments in reading and understanding written texts (including literacy and cognitive disorders). Participants were on average 31.03 ± 7.55 years old (age range 20–53) and had lived in Italy for a mean of 56 months (SD = 40.29, range 4–168 months). Males represented 95.0% of the sample (*n* = 38), 75.0% (*n* = 30) were single, and 67.5% (*n* = 27) were from Nigeria (followed by Gambia, 12.5%, and Togo, 5%). Almost half had repeatedly applied for protection visas (47.5%, *n* = 19), 5% were asylum seekers due to rules related to the Dublin Regulation. The majority had no job (67.5%) and were lacking family support in Italy (87.5%). Regarding schooling, years of study in their country were 9.87 ± 4.97. Finally, their proficiency in the Italian language was rather poor (27.5% poor knowledge of Italian language; 52.5% no knowledge at all). The two groups were similar in terms of socio-demographic variables, with very few differences: asylum seekers with PTSD were less unemployed (60% vs. 75%) and more proficient in Italian (25% vs. 15%), but with less family support (90% vs. 85%).

### 2.2. Data Collection

The sample was randomly assigned by triage nurses who referred patients upon admission. The selection was carried out on a voluntary basis: after explaining the objectives and timeframe to the patients, those who agreed to participate voluntarily were selected and informed consent was signed. To ensure correct understanding of the information, a cultural mediator was available in the case of patients with low literacy levels or questionable reading skills. Those patients who accepted to enter the study were requested to complete the research questionnaires. The self-evaluated questionnaires were administered in English or French depending on the person’s linguistic preference. If necessary, for example in case patients had reading and writing difficulties or there were cultural barriers, a cultural mediator was available to assist the patient in understanding the purpose of the research and the contents of the administered instruments. 

### 2.3. Measures

#### 2.3.1. PCL-5: PTSD Checklist for DSM-5 

The PCL-5 [14] is a 20-item questionnaire for the diagnosis of PTSD according to the DSM-5 [15]. Individuals are asked to indicate trauma-related symptoms present in the past 30 days (e.g., repeated, disturbing, and unwanted memories of the stressful experience; suddenly feeling or acting as if the stressful experience were actually happening again; feeling jumpy or easily startled, etc.). Each item is rated on a 5-point Likert scale from 0 (not at all) to 4 (extremely). The cut-off score of 33 is used for diagnosing PTSD.

#### 2.3.2. BSI-21: Bradford Somatic Inventory 

The BSI-21 [16] is a 21-item questionnaire designed to measure psychological distress expressed in somatic terms (somatization). It includes questions about a wide range of somatic symptoms (e.g., headaches, choking sensations in the throat, fluttering or feeling of something moving in the stomach, pain all over the body, etc.) and asks the subjects whether they experienced a given symptom on more or fewer than 15 days in the past month. Possible answers range from ‘absent’ (score 0) to ‘present on more than 15 days in the past month’ (score 2).

#### 2.3.3. CAMSQ: The Certainty About Mental States Questionnaire 

The CAMSQ [17] is a 20-item instrument designed to assess the capability to mentalize about self and others (e.g., I understand my feelings; when I’m in a bad mood, I know the reason why; I know how other people will react to something; I know how a person feels when I look at their face, etc.). It has two subscales, each of 10 items: the other-certainty and self-certainty scales. Each item is rated on a 7-point Likert scale from 1 (never) to 7 (always).

### 2.4. Analysis

Data were reported as mean ± standard deviation. The *t*-test and Mann–Whitney test, as appropriate, were performed to explore differences between the mean scores of the two groups (with and without PTSD) on somatization and mentalizing capacity (self and other-oriented). Pearson correlations and multiple regression analyses were conducted to examine the overall association between variables. More specifically, multivariate linear regression analyses were conducted to examine the association between self-certainty scales, PTSD and BSI. The level of significance for all analyses was set at *p* < 0.05.

## 3. Results

As shown in Table 1, asylum seekers with PTSD had significantly higher scores on somatization (*p* = 0.03) and lower scores on the self-certainty scale of mentalization (*p* = 0.04). The associations between the different variables are presented in Table 2. PTSD was positive and significantly linked with somatization (r = 0.38, *p* < 0.05). As regards mentalization, there was only a significant and negative correlation between PTSD and self-oriented mentalization (r = −0.47, *p* < 0.001). To better explore the nature and strength of the latter association a multiple linear regression was performed. The self-certainty dimension was considered as the dependent variable. This model revealed that only PTSD was significantly associated (Table 3).

## 4. Discussion

To our knowledge, this is the first study of mentalization in asylum seekers. The main finding of this study is that asylum seekers with PTSD present a reduced mentalization, which is significant in the self-certainty dimension and remains significant after controlling for the possible confounding effect of somatization.

*Self-Oriented Mentalization and Alexithymia*: The reduction in self-oriented mentalization aligns with constructs such as alexithymia, commonly observed in individuals with PTSD [18]. Alexithymia, defined as the difficulty in identifying and describing internal emotional states, is often conceptualized as a defense mechanism in the face of overwhelming emotional experiences. Considering this, our finding is in line with Söndergaard & Theorell’s [19] report of high scores on the Toronto Alexithymia Scale (TAS-20) in a group of refugees with PTSD, particularly involving the difficulties identifying own feelings. 

*Emotional Numbing and Mentalization Deficits*: The observed deficit may also be understood through the lens of emotional numbing, a core feature of PTSD [15], which includes diminished responsiveness to the external world and internal affective experience. This detachment, often adaptive in life-threatening contexts, may become maladaptive in post-trauma life, with increased rates of symptom nonimprovement, mental health problems (including substance use, suicidality, and aggressivity), poor relationship functioning, increased service utilization, and reduced quality of life [20]. Impaired self-mentalization can thus contribute to difficulties in constructing coherent self-narratives, a process crucial for trauma recovery and integration [21]. Further studies should explore whether reduced self-oriented mentalization in asylum seekers may also be related to other dimensions of post-traumatic symptoms.

*Other-Oriented Mentalization and Empathy* Reduced mentalization about others has to do with recognizing emotions, intentions, and cognitions of other people, an area usually covered by the concept of empathy. In theory, a reduced other-oriented mentalization could be expected in patients having suffered from severe interpersonal traumas, which could led to interpersonal mistrust and consequent reduced empathy. However, our findings suggest that the reduction in this dimension is not significant, in line with previous research on empathic capabilities in refugees with PTSD, which were found generally to be preserved [22]. Future studies are needed to understand whether finer dimensions of empathy are involved, but have not emerged due to the generality of the questionnaires used so far.

*Somatization*: The fact that somatization was not significantly related to a mentalization deficit was an unexpected finding, as somatization is typically the result of a mentalization deficit (i.e., people unable to make sense of their emotional experiences tend to project them onto somatically experienced symptoms). For this reason, we included somatization in this study as a potential confounder. It is too early to discuss this unexpected finding in detail, as the small sample size could influence the results. However, if confirmed by further studies, potential interpretive hypotheses will need to be developed to make sense of it.

*Possible implications for clinical practice*: A recent meta-analysis reported a consistent, large deficit in mentalizing in individuals with PTSD relative to trauma-exposed and healthy controls [3]. The same study suggests that premorbid mentalizing deficits increase the risk of PTSD. Another study found that in patients with PTSD, higher mentalizing capacities were positively correlated with higher resilience scores and lower indices of mental health [23]. In general, the ability to mentalize can play a crucial role in the treatment and management of PTSD, especially for those who have experienced complex traumatic events. Improving mentalization skills could reduce PTSD symptoms by supporting individuals to process traumatic memories, regulate overwhelming emotions, and rebuild meaningful social connections, which are often disrupted by trauma. Improving mentalization can also facilitate the process of psychological and social recovery. Indeed, psychological recovery from trauma involves emotional healing, reconstruction of a coherent self-narrative, and the rebuilding of trust in oneself and others. Mentalization plays a central role in these processes by helping individuals reflect on their internal states and those of others, leading to more adaptive coping strategies and fostering a more resilient self-concept [24]. Accordingly, mentalization-based treatments have been developed to mitigate symptoms that arise post-trauma, such as hyperarousal, hypervigilance, intrusions, flashbacks, avoidance behaviors, dissociative experiences, negative perceptions of self and others, and ensuing relational difficulties [25]. More generally, working on mentalization could help trauma survivors develop a clearer understanding of their emotional experiences, reducing rumination and facilitating the development of more adaptive coping strategies. In turn, this process could promote post-traumatic growth, which is associated with increased psychological resilience and a greater sense of personal empowerment. Although mentalization-based treatments for PTSD are typically applied to populations other than the one studied in this research, our findings confirm a mentalization problem also in asylum seekers with PTSD. Furthermore, our study shows that in asylum seekers it is the self-oriented component of mentalization that is particularly affected, which suggests that in this group of asylum seekers, psychotherapeutic techniques should focus primarily on this.

*Study limitations*: The limitations of the study include the following: (a) its exploratory nature with a limited number of subjects involved, such that the results need to be confirmed in future research on larger samples; (b) the fact that asylum seekers were mainly African males, thus limiting the possibility to generalize the results to all refugees and asylum seekers; (c) the small sample and the high prevalence of men in the sample prevent the analysis of variables such as gender or age differences, therefore further studies are needed to address these points; (d) the use of a selection of instruments to assess clinical and mentalization variables (the results could be different with other questionnaires); (e) the selection of only some variables, which hampers the evaluation of the influence of possible interfering factors related to migratory social conditions and cultural issues. Despite these limitations, we believe this study is important because it fills a gap by providing information in an area (the role of mentalization in traumatized refugees and asylum seekers) not yet covered by researchers.

## 5. Conclusions

Understanding the mental health needs of asylum seekers with PTSD is crucial in providing adequate care and support to promote their recovery and successful integration. Among other factors, the ability to mentalize may be crucial in explaining responses to trauma and has strong clinical relevance, as strengthening mentalizing skills is the target of many treatment programs. By enhancing the ability to reflect on and understand their own mental states, asylum seekers with PTSD can develop healthier emotional regulation, a more coherent self-concept, and better social connections. These processes are essential for overcoming trauma and rebuilding a fulfilling life. Accordingly, if further studies confirm our exploratory findings, treatment programs for asylum seekers could be designed to include the assessment and enhancement of mentalization capabilities. According to our data, this could be particularly relevant regarding the ability to experience and recognize one’s feelings, wishes, cognitions, and behaviors.

A possible protocol could be adapted to migrants from Fonagy’s “Mentalization-Based Treatment (MBT)” [26], which is not a rigid, single procedure, but rather a treatment approach that focuses on developing mentalization capacity through the therapeutic relationship, interpretation of behaviors, and the promotion of a deeper understanding of the patient’s mental states. Key concepts used in this protocol are as follows: (a) *Mentalization*: the ability to interpret one’s own and others’ actions as meaningful based on mental states (thoughts, feelings, intentions, and desires), even implicitly. (b) *Reflective Functioning*: a central metacognitive ability in Fonagy’s theory, allowing individuals to build a “theory of mind” to guide their interactions with the world. (c) *The Therapeutic Relationship*: the therapeutic environment is essential—not only as a space for analysis but as an active component of the process. In this kind of therapeutic intervention, patients are encouraged to develop a more complex theory of mind, allowing them to navigate social interactions more effectively. The steps of a possible protocol to be developed for these patients could include the following: 1. Evaluation of mentalization through a test at time 0. 2. A cycle of psychotherapy sessions aimed at processing traumas and becoming aware of one’s mentalization style, and identifying possible points of change. 3. A follow-up test assessing the effectiveness on mentalization capability as well as on psychopathological phenomena. Finally, to be applied effectively, the protocol should be adapted to both cultural differences and post-traumatic sensitivity to triggers that remind one of the traumatic experience. Focus groups including clinicians, cultural mediators, and former patients could be used in order to address the needed adaptation of the protocol to the specific context of traumatized asylum seekers. 

In conclusion, improving mentalization may not only contribute to the reduction in PTSD symptoms, but it can also facilitate the psychological and social recovery of trauma survivors.

## Figures and Tables

**Table 1 ijerph-22-01405-t001:** Descriptive statistics for asylum seekers with PTSD (*n* = 20) and without PTSD (*n* = 20).

Measure	PTSD			
	YES		NO	
	M	(SD)	M	(SD)
BSI	19.30	(7.37)	13.60	(8.32)
Other-Certainty (CAMSQ)	3.30	(0.99)	4,03	(1.74)
Self-Certainty (CAMSQ)	5.12	(1.32)	5,92	(0.97)

Abbreviations: PTSD = Post-traumatic stress disorder; BSI = Bradford Somatic Inventory; CAMSQ = Certainty About Mental States Questionnaire.

**Table 2 ijerph-22-01405-t002:** Correlational analyses between post-traumatic stress disorder, somatization, and mentalization.

	Scores	1	2	3	4
1	PCL-5	-			
2	BSI	0.38 *	-		
3	Other-certainty	−0.28	−0.13	-	
4	Self-certainty	−0.47 **	−0.22	0.54 ***	-

* *p* < 0.05, ** *p* < 0.01, *** *p* < 0.001; Abbreviations: PCL-5, PTSD Checklist for DSM-5; BSI, Bradford Somatic Inventory.

**Table 3 ijerph-22-01405-t003:** Influence of post-traumatic stress disorder and somatization on self-oriented mentalization.

Variables	β	*t*	*p*-Value
Self-certainty scale (CAMSQ)			
PCL-5	−0.03	2.86	<0.01
BSI	−0.01	0.33	0.75

Abbreviations: PCL-5, PTSD Checklist for DSM-5; BSI, Bradford Somatic Inventory; CAMSQ = Certainty About Mental States Questionnaire.

## Data Availability

The raw data supporting the conclusions of this article will be made available by the corresponding author on request.

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
