# Peer review of "Mentalization in Asylum Seekers with Post-Traumatic Stress Disorder"

_ijerph, 2025, doi:10.3390/ijerph22091405_

Round 1

Reviewer 1 Report

Comments and Suggestions for Authors

The article deal whit an important topic, the mentalization of asylum seekers with Post Traumatic Stress Disorder.

The topic is relevant, it focuses on showing evidence that helps to understand the situation and consequently propose better interventions with this group.

It is an exploratory, innovative study on how mentalization can overcome trauma and traumatic stress in asylum seekers.

In the introduction, the authors define the concept of mentalization and reveal its importance. However, it is necessary to address the relevance of mentalization to asylum seekers, because in these cases there are traumatic situations that they go through, which generate not only traumatic stress but also other mental illnesses. It would be relevant to address here the relevance of applying this process to this population, showing with evidence, the existence, or not, of studies on the subject, or similar studies, which could be useful to explain the relevance of the study. For example, there are several studies on the high prevalence of mental problems and psychopathologies in this group. I would therefore advise to carry out a brief of the literature review on the subject.

In methodological terms, it is necessary to clarify whether the asylum seekers did not speak Italian well, did they answer the items? The questionnaire was administered by interviewers or directly.

A second clarification concerns the existence of two study groups. In the discussion, authors presented the characteristics of the group as a whole, but don't differentiate them sociographically

These people are volunteers or have been selected through a sample.

Although it says at the end that there was an ethical protocol, it would be important here to clarify whether people individually consented to participate and how that consent was obtained, since people don't speak much Italian

In the results.

The tables show the data from the scales but do not mention the groups, the data is presented as a whole. Clarifying this part would be important.

It would also be relevant to show whether there are differences between the two groups and whether there are differences between men and women as well as between younger and older people.

Discussion

The discussion needs to be in-depth and detailed, taking into account the questions posed above and also according to the literature review that should be in the introduction.

The implications for practice should be written down in the form of a protocol on how to proceed in these cases to develop a process of mentalization for asylum seekers in the host countries. For example, psychological support, professionals trained to promote this type of process.

Author Response

Reply to reviewer 1

The article deal whit an important topic, the mentalization of asylum seekers with Post Traumatic Stress Disorder.

The topic is relevant, it focuses on showing evidence that helps to understand the situation and consequently propose better interventions with this group.

It is an exploratory, innovative study on how mentalization can overcome trauma and traumatic stress in asylum seekers.

In the introduction, the authors define the concept of mentalization and reveal its importance. However, it is necessary to address the relevance of mentalization to asylum seekers, because in these cases there are traumatic situations that they go through, which generate not only traumatic stress but also other mental illnesses. It would be relevant to address here the relevance of applying this process to this population, showing with evidence, the existence, or not, of studies on the subject, or similar studies, which could be useful to explain the relevance of the study. For example, there are several studies on the high prevalence of mental problems and psychopathologies in this group. I would therefore advise to carry out a brief of the literature review on the subject.

We wrote in the introduction the following sentence that is in line with this request: “Among the severely traumatized individuals who often suffer from PTSD and related post-traumatic reactions (e.g. complex PTSD) are asylum seekers, who are exposed to traumatic experiences in their countries and/or during the migration journey. However, to our knowledge research on the relationship between PTSD and mentalization in asylum seekers is lacking. Consequently, this study aim to fill this gap by exploring the relation-ship between PTSD and the two major dimensions of mentalization (self and other-oriented).” Moreover, in the discussion our data are contrasted to the rare studies that indirectly explore part of the mentalization construct (e.g., studies on empathy which roughly corresponds to the other-oriented dimension).

In methodological terms, it is necessary to clarify whether the asylum seekers did not speak Italian well, did they answer the items? The questionnaire was administered by interviewers or directly.

To clarify this point, we added in the section “data collection” the following sentence: “The self-evaluated questionnaires were administered in English or French depending on the patients’ linguistic preference”, followed by the sentence about the assistance of cultural mediators in case clarification was needed.

A second clarification concerns the existence of two study groups. In the discussion, authors presented the characteristics of the group as a whole, but don't differentiate them sociographically

The socio-demographic characteristics of the sample are presented in the section “participants”. The minor differences between the two groups are also showed in that section, with the following sentence: “The two groups were similar in terms of socio-demographic variables, with very few dif-ferences: patients with PTSD were less unemployed (60% vs. 75%) and more proficient in Italian (25% vs. 15%), but with less family support (90% vs. 85%)”

These people are volunteers or have been selected through a sample.

To clarify this point, we added the following sentence: “Patients were randomly selected from those accessing our “International Protection Service”.”

Although it says at the end that there was an ethical protocol, it would be important here to clarify whether people individually consented to participate and how that consent was obtained, since people don't speak much Italian

We thank the reviewer for this comment. We added the following sentence to address this point: “To enter the study, all patients signed an individual consent to participate, available in Italian, English and French. In case of minor linguistic difficulties, a cultural mediator was available to translate the consent to participate in the patients’ native language. Patients who were unable to read any of these languages were excluded from the study (provided that they could not read the questionnaires (see below).”

In the results.

The tables show the data from the scales but do not mention the groups, the data is presented as a whole. Clarifying this part would be important.

The two groups are contrasted in table 1. The correlation analyses and the regressions are to be performed on the entire group (tables 2 and 3).

It would also be relevant to show whether there are differences between the two groups and whether there are differences between men and women as well as between younger and older people.

We agree that it would be interesting; however, we did not perform this analysis due to the small size of the sample.

Discussion

The discussion needs to be in-depth and detailed, taking into account the questions posed above and also according to the literature review that should be in the introduction.

Compared to the previous version, the discussion was enlarged and more points were raised. We think that due to the exploratory nature of this brief report, a more large discussion would lose the focus.

The implications for practice should be written down in the form of a protocol on how to proceed in these cases to develop a process of mentalization for asylum seekers in the host countries. For example, psychological support, professionals trained to promote this type of process.

The implications of the study are addressed in the conclusions, there it is stressed that “Understanding the mental health needs of asylum seekers with PTSD is crucial in providing adequate care and support to promote their recovery and successful integration. Among other factors, the ability to mentalize may be crucial in explaining responses to trauma and has strong clinical relevance, as strengthening mentalizing skills is the target of many treatment programs. By enhancing the ability to reflect on and understand their own mental states, asylum seekers with PTSD can develop healthier emotional regulation, a more coherent self-concept, and better social connections. These processes are essential for overcoming trauma and rebuilding a fulfilling life.”

We also included practical recommendations: “…if further studies confirm our exploratory findings, treatment programs for asylum seekers could be designed to include the assessment and enhancement of mentalization capabilities. According to our data, this could be particularly relevant regarding the ability to experience and recognize one’s feelings, wishes, cognitions and behaviors. More in detail, a protocol could include: 1. Evaluation of mentalization through a test at time 0. 2. A cycle of psychotherapy sessions aimed at processing traumas and becoming aware of one’s mentalization style, and identifying possible points of change. 3. Mentalization follow-up test. In conclusion, improving mentalization may not only contribute to the reduction of PTSD symptoms, but it can also facilitates the psychological and social recovery of trauma survivors.”

Reviewer 2 Report

Comments and Suggestions for Authors

Dear Authors,

Your brief report is an important piece of research and fills a gap in the existing knowledge base. My comments pertain to the abstract and references; otherwise, the report is well-written.

Lines 7–9: Please include the motivation for the study here, drawing from lines 22–23, e.g., “This study is important because…”

Lines 7–9: Please add some information about the participants, including when the data was collected and the sample size (N).

Lines 11, 17, 20, 24: Please ensure consistency regarding the description of participants: are they asylum seekers, patients, or both?

Lines 99–102: Please include some examples of the questions asked in this section.

Lines 112–114: Please provide some examples here as well.

Line 171: Why is the port explosion mentioned here? It appears out of context; please elaborate to clarify its relevance.

Lines 235–292: Please ensure consistent formatting for all references. The current reference list contains several different citation styles.

Author Response

Reply to reviewer 2

Dear Authors,

Your brief report is an important piece of research and fills a gap in the existing knowledge base. My comments pertain to the abstract and references; otherwise, the report is well-written.

Thank you for this comment

Lines 7–9: Please include the motivation for the study here, drawing from lines 22–23, e.g., “This study is important because…”

Done

Lines 7–9: Please add some information about the participants, including when the data was collected and the sample size (N).

We inserted here the number of participants, the information about the year of evaluation has been added in the section “participants”

Lines 11, 17, 20, 24: Please ensure consistency regarding the description of participants: are they asylum seekers, patients, or both?

Thank you for noticing this discrepancy. Working in a healthcare system, we tend to think of everyone as patients, but in reality, it's more accurate to simply call them asylum seekers. We've revised the manuscript to remove the term "patient" where appropriate.

Lines 99–102: Please include some examples of the questions asked in this section.
Done

Lines 112–114: Please provide some examples here as well.
Done

Line 171: Why is the port explosion mentioned here? It appears out of context; please elaborate to clarify its relevance.

Lacking studies addressing the role of mentalization in PTSD in asylum seekers, one of the few available studies is the one concerning the role of mentalization in PTSD in people facing the traumatic experience of the port explosion. We reframed that sentence to consider the reviewer’s comment.

Lines 235–292: Please ensure consistent formatting for all references. The current reference list contains several different citation styles.

Done